# Two-Dimensional (2D) TM-Tetrahydroxyquinone Metal–Organic Framework for Selective CO_2_ Electrocatalysis: A DFT Investigation

**DOI:** 10.3390/nano12224049

**Published:** 2022-11-17

**Authors:** Xianshi Zeng, Chuncai Xiao, Luliang Liao, Zongxing Tu, Zhangli Lai, Kai Xiong, Yufeng Wen

**Affiliations:** 1School of Mathematical Sciences and Physics, Jinggangshan University, Ji’an 343009, China; 2Institute for Advanced Study, School of Physics and Materials Science, Nanchang University, Nanchang 330031, China; 3School of Mechanical and Electrical Engineering, Xinyu University, Xinyu 338004, China; 4School of Chemistry and Chemical Engineering, Nanchang University, Nanchang 330031, China; 5Materials Genome Institute, National Center for International Research on Photoelectric and Energy Materials, School of Materials and Energy, Yunnan University, Kunming 650091, China; 6Advanced Computing Center, Information Technology Center, Yunnan University, Kunming 650091, China

**Keywords:** electrocatalytic CO_2_ reductionreaction, MOF, TM tetrahydroxyquinone, density functional theory (DFT) calculations

## Abstract

The resource utilization of CO${}_{2}$ is one of the essential avenues to realize the goal of “double carbon”. The metal–organic framework (MOF) has shown promising applications in CO${}_{2}$ catalytic reduction reactions due to its sufficient pore structure, abundant active sites and functionalizability. In this paper, we investigated the electrocatalytic carbon dioxide reduction reactions of single-atom catalysts created by MOF two-dimensional coordination network materials constructed from transition metal-tetrahydroxybenzoquinone using density function theory calculations. The results indicate that for 10 transition metals, TM-THQ single levels ranging from Sc to Zn, the metal atom binding energy to the THQ is large enough to allow the metal atoms to be stably dispersed in the THQ monolayer. The Ni-THQ catalyst does not compete with the HER reaction in an electrocatalytic CO${}_{2}$ reduction. The primary product of reduction for Sc-THQ is HCOOH, but the major product of Co-THQ is HCHO. The main product of Cu-THQ is CO, while the main product of six catalysts, Ti, V, Cr, Mn, Fe, and Zn, is CH${}_{4}$. The limit potential and overpotential of Ti-THQ are the highest, 1.043 V and 1.212 V, respectively. The overpotentials of the other monolayer catalysts ranged from 0.172 to 0.952 V, and they were all relatively low. Therefore, we forecast that the TM-HQ monolayer will show powerful activity in electrocatalytic carbon dioxide reduction, making it a prospective electrocatalyst for carbon dioxide reduction.

## 1. Introduction

From the industrial revolution onwards, the excessive use of fossil fuels has resulted in a significant rise in the concentration of carbon dioxide (CO${}_{2}$) in the atmospheres, which could lead to devastating outcomes for the modern world. The conversion of carbon dioxide into useful chemical raw materials can help to ease environmental issues including the “greenhouse effect” and conventional methods’ reliance on fossil raw materials such as crude oil [1,2,3]. Therefore, CO${}_{2}$-based conversion reactions have received a lot of attention. CO${}_{2}$ reduction reaction is one of the typical representatives, which can directly obtain C${}_{1}$-C${}_{3}$ high value-added chemical products (such as CO, CH${}_{4}$, HCOOH, HCOH, CH${}_{3}$OH, C${}_{2}$H${}_{4}$, C${}_{2}$H${}_{6}$, C${}_{2}$H${}_{5}$OH, C${}_{3}$H${}_{6}$, etc.) through electrochemistry [4,5], chemical reforming [6], photochemistry [7], biochemistry [8], etc. Among them, electrocatalytic reaction conditions are milder, and the process is the high cleanliness of the process is of great practical importance. However, this technology also faces the problems of low catalytic efficiency, limited yield and insufficient catalyst life, so there is an urgent need to devise and fabricate new catalysts with high efficiencies and low overheads.

A crystal structure known as a metal–organic framework (MOF) is made up of inorganic metal ions and organic ligands. It can be made using a bottom–up assembly method under solvothermal conditions and has the advantages of having an extremely diverse structural makeup, being highly porous, and having a high specific surface area [9,10,11,12,13,14], thus showing excellent performance in sensing, separation, energy storage, and other fields, especially in the field of catalysis [15,16,17,18,19,20,21,22,23,24]. MOFs have the following advantages over conventional catalytic materials: (1) The porous characteristics provide a rich particular surface area for reaction, which promoted the adsorption of reactants. (2) The confinement of pore channels produces effects such as spatial and electron limitation, optimizing reaction intermediates and selectivity [25]. (3) The metal centers, as catalytic active sites, can be uniformly distributed on the internal and external surfaces of the material at the atomic level. This usually results in higher catalytic efficiency than conventional catalysts with active sites concentrated on the external surface. (4) The metal centers and ligands are matched electronically to form a characteristic energy band structure, and some MOFs have certain electrical conductivity, which is suitable for the absorption and transmission of electrical energy. (5) The well-defined crystalline configuration of MOF can help to study the reaction mechanism and establish the correspondence between structure and catalytic performance. In addition, the high affinity of MOF for CO${}_{2}$ greatly enhances the correlation between reactants and catalysts, which promotes the efficient reaction. Therefore, MOF has become one of the popular materials at the frontier of CO${}_{2}$ electrocatalytic reduction. In recent years, several researchers have conducted extensive work on MOF electrocatalytic CO${}_{2}$ reduction. For instance, Zhu et al. [26] presented a strategic approach for the in situ electrosynthesis of hollow copper MOF and its reduction to a multistage Cu dendrimer catalyst in a reaction in which the Cu-MOF film was prepared in 5 min. The Cu dendrimer catalyst obtained by this strategy has a high surface area with an open reactive site, which is favorable for the reduction of CO${}_{2}$ to formate. In the electrolyte of the ionic liquid, the current density of the system was up to 102.1 mA/cm${}^{2}$ and the selectivity was 98.2%. Kang et al. [27] used copper foam as a substrate to control the MOF growth kinetics by using a ligand with high spatial site resistance, resulting in the rapid growth of Cu-MOF with a large number of defects on copper foam. The catalyst was able to reduce CO${}_{2}$ to formate in an acetonitrile ionic liquid electrolyte with a Faraday efficiency of 90.5%. In addition, other researchers have constructed MOFs by ligand doping, which have also shown excellent performance for electrocatalytic CO${}_{2}$ reduction [28,29]. More interestingly, conductive materials facilitate charge transport and are more advantageous in electrocatalysis. In a mixed electrolyte of 1 mol/L choline chloride and 1 mol/L potassium hydroxide, Majidi et al. [30] created a copper-based, conductive, two-dimensional MOF: copper-tetrahydroxyquinone (Cu-THQ) nanosheets, which required only a low overpotential of 16 mV for CO${}_{2}$ reduction. The catalytic activity was excellent. The current density was around 173 mA/cm${}^{2}$, and the average CO Faraday efficiency was approximately 91% at an applied voltage of −0.45 V (with respect to the reversible hydrogen electrode). Their great progress in the laboratory inspired us to explore the application of TM-THQ electrocatalytic CO${}_{2}$ reduction in depth. Here, we have constructed a series of TM-THQ metal–organic framework materials with 3D transition metals and conducted a systematic study by DFT in CO${}_{2}$ electrocatalytic reduction reactions, where computational simulations are performed to screen possible pathways for cost-effective novel catalysts.

## 2. Details of The Calculation

Using spin-polarized density generalized function theory (DFT), the Dmol3 package was used for all computations in this study [31]. Under the generalized gradient approximation, the Perdew–Burke–Ernzerhof (PBE) function describes the exchange correlations between electrons (GGA) [32]. The DNP basis group is used in conjunction with the effective potential DSPP approximation to address the basis group, which substitutes the kernel electron with a single effective potential and incorporates relativistic corrections [33]. A van der Waals correction (DFT-D2) was included to the computations to better explain the adsorption of molecules on the surface due to the weak long-range interactions between the layers [34,35,36,37]. The conductor approximation shielding model (COSMO) was used as the solventization model since the real catalytic process takes place in an aqueous environment. This choice improved agreement with the experiment. The water (relative permittivity ϵ = 78.54) was used as the solvent to simulate the effect of the solvent on all systems [38]. The interaction between adjacent heterogeneous joints was eliminated by selecting a vacuum layer thickness of 25. The Monkhorst–Pack method [39] K-point grid with 2 × 2 × 1 and 6 × 6 × 1 are used for the structural determination and electronic structure calculations, respectively. To increase the computational accuracy, the energy convergence criterion is 10${}^{-6}$ eV.

For the purpose of discussing the strength of the interactions between intermediates and THQ monolayers, Equation (Equation 1) provides a defined energy of adsorption (Eads) (as exemplified by the adsorption energy of CH${}_{4}$).
(1)$$E_{ads}=E_{TM-THQ-CH_{4}}-E_{TM-THQ}-E_{CH_{4}}$$
where $E_{TM-THQ-CH_{4}}$, $E_{TM-THQ}$ and $E_{CH4}$ denote the sum energy of CH${}_{4}$ that is adsorbed on the surface of TM-THQ, the sum energy of the TM-THQ unilayer and the sum energy of individual CH**${}_{4}$** molecules, respectively. Since CO${}_{2}$RR is characterized by a large number of reaction paths, the notion of Gibbs free energy was introduced to enable the determination of the most suitable reaction path. When the reaction involves the transfer of electrons, the energy of the reaction can be calculated by the standard electrode model for hydrogen presented by Nørskov and coworkers [40,41,42]. Equation (Equation 2) is the equation for the Gibbs free energy.
(2)$$\Delta G=\Delta E+\Delta E_{ZPE}-T\Delta S+\Delta G_{pH}+\Delta G_{U}$$
where ΔE is a reaction energy of the reaction, $\Delta E_{ZPE}$ and ΔS is the change in zero-point energies and entropy values. Here, *T* is the reaction’s thermodynamic temperature (298.15 K). $\Delta G_{pH}$ is the free energy correction ($\Delta G_{pH}=2.303k_{B}TpH$) introduced by the different acidity of the solution (different H+ concentrations), assuming a pH of 0 in acidic solutions. While $\Delta G_{U}$ is the free energy correction brought about by the various electrode potentials, $\Delta G_{U}$ may be calculated using Equation (Equation 3) as follows: (3)$$\Delta G_{U}=-neU$$
where *U* is the applied electrode potential and *n* is the total number of transferred electrons. When determining if the catalyst has great catalytic activity, the limiting potential ($U_{L}$) and overpotential (η) are crucial considerations. The limiting potential is calculated from Equation (Equation 4).
(4)$$U_{L}=-\Delta G_{max}/ne$$

$\Delta G_{max}$ is the value of the change in the free energy of the decisive step. The over-potential can be derived by computing the differential between the equilibrium potential (Uequilibrium) and the limiting potential and the limiting potential obtained as the difference shown in Equation (Equation 5).
(5)$$\eta =U_{equilibrium}-U_{L}$$

## 3. Results and Discussion

### 3.1. The Structural Characteristics and Features of TM-THQ Monolayer

The structure of the MOF material constructed by TM-THQ is shown in Figure 1. Here, (a) and (b) are, respectively, the top and side views of a single cell, and (c) is a top view of a 2 × 2 supercell. The specific lattice constant values of these ten MOF materials after optimization are presented in Appendix A. The top view of the monocell clearly shows that each monocell contains 12 C atoms, 12 O atoms, and 3 transition metal atoms, and that each metal atom is coordinated to 4 O atoms in the tetrahydroxybenzoquinone. Each atom of the 10 transition metals (from Sc to Zn) we considered resides in the same plane (Figure 1). The bond lengths between metal atoms and their nearest O atoms are in the range of 1.858 to 2.066 Å (Figure 2). The bond length of Sc-O is 2.066 Å, which is consistent with the fact that Sc has the largest atomic radius among the 10 metals. Furthermore, in order to study the electronic state of the monolayer, we performed a Hirshfeld charge analysis, and it can be seen from Table 1 that all the metal atoms in the 10 metals of the first transition metal series considered have a partial positive charge, while the nearest O atom has an opposite negative charge, which indicates that the metal atoms transfer some of their electrons to the THQ monolayer, making the metal atoms and O atoms interact with each other as ionic bonds in addition to ligand bonds. The analysis of the spin states of the metal atoms shows that Cr exhibits a spin state with a magnetic moment of 2.977 $\mu_{B}$, while the rest of the metal atoms are in the non-spin state.

### 3.2. The Stabilization of TM-HQ Single Layer

The stability is an essential parameter for evaluate the catalytic performance of catalysts. To investigate the stability of the TM-THQ monolayer, we have calculated the cohesion values of the bulk metal and the binding values of the TM-THQ monolayer. The cohesion energy can be obtained by the equation: $E_{c}=\left(E_{TM(bulk)}-nE_{TM}\right)/n$, where $E_{M(bulk)}$ and $E_{TM}$ are the energies of the bulk and individual metal atoms, respectively. The number of *n* is the number of metal atoms in the volume. The binding energy is calculated by the equation: $E_{b}=E_{TM-Pc}-E_{TM}-E_{Pc}$, where $E_{TM-Pc}$, $E_{TM}$ and $E_{Pc}$ are the energies of TM-Pc monolayer, individual metal atoms and phthalocyanine monolayer, respectively. For single-atom accelerators, the stronger the binding ability between the metal atoms and the substrate, the more difficult it is for the metal atoms to agglomerate with each other; this results in a uniform and stable mosaic of metal atoms on the substrate. Appendix A gives the binding energy and the cohesion energy of the corresponding metals for these ten TM-THQ catalysts. The cohesion energies of the bulk metal parts are in the range of −6.578 to −1.054 eV, while the binding energies in the case of transition metal atoms and tetrahydroxyquinone monolayer are in the range of −11.045 to −4.628 eV.

It is obvious from Figure 3 that the cohesion energy of the metal bulk is below the binding energy of the corresponding metal TM-HTQ monolayer, which indicates that the metal atoms tend to bind to the tetrahydroxyquinone monolayer more than the metal atom agglomeration, thus making the metal atoms uniformly mosaic into the cavities of the tetrahydroxyquinone monolayer, and thus, the TM-HTQ monolayer has good stability. Meanwhile, the binding energy between these 10 transition metals (from Sc to Zn) and the tetrahydroxybenzoquinone monolayer shows a gradual increasing trend and weakening bond solidity. This is because the oxygen atom is a strong non-metallic atom, while Sc to Zn are of the same period, and the metallicity gradually decreases. Generally speaking, the stronger the metallicity, the stronger the bonding with the surrounding oxygen atoms, and conversely, the weaker the metallicity, the weaker the bonding with the oxygen atoms, thus showing a gradual increase in the bonding energy. The reason for the irregular fluctuations of the cohesion energy of the ten metals is that the ten transition metals belong to different crystal systems, among which Sc, Ti, Co, and Zn belong to a hexagonal structure; V, Cr, Fe, Ni, and Cu belong to a cubic crystal system structure, and Mn belongs to a trigonal crystal system structure. Different crystal systems have different atomic arrangements, which affect different ground state energies and thus exhibit irregular fluctuations.

### 3.3. TM-THQ Single Layer for CO${}_{2}$RR and HER Specificity

Generally, electrocatalytic CO${}_{2}$ reduction takes place in solution as a multiple electron step process. The proton–electron pairs (H^+^ + e^−^) in solution are increasingly involved in the reaction under the action of external voltage, and after the molecules of carbon dioxide are adsorbed on the surface of the catalyst, two separate intermediates are produced during the first protonation step, depending on the position of the added H. If H is included in the oxygen atom (* + CO_2_ + H^+^ + e^−^ → *COOH ), the intermediate *COOH will be obtained. Conversely, if H is included in the carbon atom (* + CO_2_ + H^+^ + e^−^ → *OCHO), the intermediate *OCHO will be developed. However, H can be also added to the metal atom in the catalyst (* + H^+^ + e^−^ → *H), resulting in a hydrogen precipitation reaction (HER), which is undesirable for CO${}_{2}$RR. In fact, the two reactions, CO${}_{2}$RR and HER, are in competition, so for CO${}_{2}$RR catalysts, we have to consider the material selectivity for CO${}_{2}$RR and HER.

Figure 4 compares the variations of *OCHO, *COOH and *H Gibbs free energies formed by the first-step protonation reaction. As shown in Figure 4, without considering the HER reaction, the first transition metal series of Mn, Fe, Co, Ni, and Cu are more likely to form intermediates *COOH, while Cr, V, Ti, Sc, and Zn are more likely to form intermediates *OCHO. When considering the competition with HER, it is obvious that for both Cu-THQ and Zn-THQ catalysts, the Gibbs free energy is lower than that of *H, regardless of whether *COOH or *OCHO intermediates are formed, showing a good inhibition of HER reaction and good electrocatalytic CO${}_{2}$ reduction activity. For Ni-THQ catalysts, the formation energy is higher than that of *H, regardless of whether *COOH or *OCHO intermediates are formed, which indicates that once the active site receives protons to generate *H, mainly hydrogen precipitation reaction occurs, and thus, it can be concluded that Ni-THQ does not compete with the HER reaction for CO${}_{2}$ electrocatalysis.

In addition, from Figure 4, it can be seen that the Gibbs free energy formed of *OCHO intermediates is higher than that for the formation of *H for the metals Mn, Fe, and Co, but the Gibbs free energy for the generation of *COOH intermediates is lower than that for the generation of *H. For the metals V, Ti, and Sc, the formation energy for the obtained of *COOH intermediates is also higher than that for the obtained of *H, but the production of *OCHO intermediates with Gibbs free energy is lower than at the time for the production of *H. Once the active site of the metal is occupied by O*CHO or C*OOH, there is no excess active site to accept H*. Therefore, it can be concluded that the six catalysts Mn, Fe, Co, V, Ti, and Sc are also catalytically active for the CO${}_{2}$RR reaction.

### 3.4. Potential Product Routes and Adsorbed Energy

Owing to TM-THQ electrocatalytic CO${}_{2}$ reduction is a single-atom catalytic procedure, it is widely assumed that it is difficult to produce poly-carbon products because the monoatomic catalytic process cannot achieve C–C coupling between intermediates. Therefore, theoretically, the single-atom catalytic CO${}_{2}$ reduction process is predicted as long as the C${}_{1}$ productos is considered. The most common C${}_{1}$ products in the electrocatalytic reduction of carbon dioxide are CO, CH${}_{4}$, HCOOH, CH**${}_{3}$**OH, and HCHO. Figure 4 shows the scheme of electrocatalytic reduction of CO${}_{2}$ to obtain C${}_{1}$ products [43].

As can be seen in Figure 5, the reduction of CO${}_{2}$ to produce CO, HCOOH is a 2e process. The reduction paths are CO_2_ → *COOH → *CO → CO and CO_2_ → *OCHO → *OCHOH → HCOOH. The generation of HCHO is a 4e process, and the reduction paths are CO_2_ → *COOH → *CO → *CHO → *OCH_2_ → HCHO. The acquisition of CH${}_{3}$OH products is a 6e process, and the reduction paths are CO_2_ → *COOH →*CO → *CHO → *OCH_2_ → *OCH_3_ → *OHCH_3_ → CH_3_OH. The most complicated is the obtaining of the CH_4_ product, which is an 8e process with three possible paths, namely: (1) CO_2_ → *COOH → *CO → *COH → *C → *CH → *CH2 → *CH_3_ →* + CH_4_; (2) CO_2_ → *COOH → *CO → *CHO → *OCH_2_ → *OCH_3_ → *OHCH_3_ → *O + CH_4_ → *OH → H_2_O and (3) CO_2_ → *COOH → *CO → *CHO → *OCH_2_ → *OCH_3_ → *OHCH_3_ → *OH + CH_4_ → * + H${}_{2}$O.

In order to predict the most likely product for each catalyst based on the complexity of the CO${}_{2}$ electrocatalytic reduction reaction pathway, we first calculated the adsorption energy of the catalyst for the C${}_{1}$ product, as shown in Table 2.

As can be seen in Table 2, Sc, Co as well as Cu have a weak adsorption capacity for all five C${}_{1}$ products and can be obtained by smooth desorption from the catalyst surface once the products are generated. Although Ni exhibits the same weak adsorption capacity for the five C${}_{1}$ products, it is mainly manifested as an HER reaction process according to Figure 3 and thus is not considered. For the three single-atom catalysts, Ti, V and Cr, they have a strong adsorption capacity for CO, HCOOH, HCHO and CH${}_{3}$OH, and therefore, these products are firmly adsorbed in the catalysts and cannot be desorbed, leading to the poisoning of the catalytic process to obtain the products. Fortunately, the adsorption capacity for CH${}_{4}$ is not strong, thus making it possible to obtain the CH${}_{4}$ products. The C${}_{1}$ products that can be desorbed from the surface of Mn-THQ are HCOOH, HCHO, and CH${}_{4}$. The Zn-THQ catalyst is strongly adsorbed for CH${}_{3}$OH, and this product cannot be obtained, but the other four C${}_{1}$ products are not strongly adsorbed, and thus, CO, HCOOH, HCHO, and CH${}_{4}$ can be obtained.

### 3.5. The Reaction Route of Electro-Chemical Reduction of Carbon Dioxide

#### 3.5.1. HCOOH as the Main Catalytic Product

According to the scheme in Figure 5, the first-step protonation of CO${}_{2}$ after the adsorption of Sc-THQ occurs in the presence of external electricity potential to form *COOH or *OCHO intermediates. However, according to Figure 4, it can be seen that the Sc-HQ catalyst requires crossing a relatively high energy bar for the formation of *COOH intermediates in the first step of protonation, and the competitive reaction with HER shows weakness; thus, we only consider the generation of an *OCHO intermediate step in the electrocatalytic CO${}_{2}$ reduction process.

As can be seen from Figure 6, the generation of *OCHO intermediate is a Gibbs free energy drop process and the reaction is easy to proceed, and the second paper protonation step occurring on this basis to generate *OCHOH intermediate is also a free energy drop process. Then, after the formation of *OCHOH, this intermediate has the potential to undergo a *OCHOH → H^+^ + e^−^) → *CHO/*OCH + H_2_O protonation reaction, but it needs to cross the energy barrier of 1.289/1.677 eV. In contrast, the direct desorption of *OCHOH to form HCOOH requires crossing only an energy barrier of 0.816 eV. The detailed values of the free energy are given in Appendix A. Thus, HCOOH desorption prevails overwhelmingly, and the product is obtained while the reduction reaction is terminated. The reaction path of the whole process is * + CO_2_ → *OCHO →*OCHOH → HCOOH where the rate-determining step is *OCHOH → * + HCOOH and the limiting potential is 0.816 V.

#### 3.5.2. HCHO Is the Main Catalytic Product

We found that for the Co-THQ catalyst, the main product is HCHO. Figure 7 shows the step diagram for each step of Co-THQ electrocatalytic CO${}_{2}$ reduction. Detailed values of the Gibbs free energy change at every protonation step are displayed in Appendix A. According to Figure 4, the Co-THQ catalyst needs to require crossing a high energy bar in the first step of protonation to form a *OCHO intermediate, and the competitive reaction with HER shows weakness; thus, we only consider the step of generating *COOH intermediate in the electrocatalytic CO${}_{2}$ reduction process. As can be seen from Figure 7, after the adsorption of CO${}_{2}$ molecules onto the surface of Co-THQ, the first step of protonation to form *COOH is a free energy rise process, which requires overcoming an energy barrier of 0.242 eV. The second step of protonation to form *CO intermediates is also a free energy rise process, but the energy barrier to be overcome is relatively small, with a value of 0.179 eV. After the formation of *CO intermediates, further protonation to form *CHO/*COH is possible, and CO desorption may also occur to obtain CO products. However, according to Appendix A, it can be seen that the formation of *COH as well as CO desorption requires overcoming a higher energy barrier of 1.294 eV and 0.528 eV, respectively, while the formation of *CHO is a free energy drop process. Thus, we consider this step as the formation of *CHO. Further protonation to form *OCH${}_{2}$ is also an exothermic reaction with a free energy drop. After the formation of *OCH${}_{2}$ intermediate, further protonation to form *OCH${}_{3}$ requires crossing a higher energy barrier of 0.533 eV, while HCHO desorption only requires overcoming an energy barrier of 0.196 eV. Therefore, HCHO desorption occurs at this step to obtain the product and terminate the reaction. The whole process of HCHO product generation is * + CO_2_ → *COOH → *CO → *CHO → *OCH_2_ → HCHO. The rate-determining step is * + CO_2_ + H^+^ + e^−^) → *COOH, and the corresponding limiting potential is 0.242 V.

#### 3.5.3. With CH${}_{4}$ as the Main Catalytic Product

From the adsorption energies in Table 2, it is clear that the MOF materials constructed with three atoms of Ti, V and Cr and tetrahydroxybenzoquinone have strong adsorption of CO, HCOOH, HCHO and CH${}_{3}$OH, and the products cannot be obtained by desorption. When these three metal-tetrahydroxybenzoquinones were used as electrocatalytic CO${}_{2}$ reduction, only CH${}_{4}$ products were obtained. Gibbs free energy calculations after the protonation step showed that the main products obtained by the electrocatalytic CO${}_{2}$ reduction of Mn, Fe, and Zn atoms were also CH${}_{4}$. The free energy step curves for each protonation step of these six catalysts are given in Figure 8.

From Figure 4, it can be seen that the energy barrier to be crossed in the first step of protonation to form *COOH in the electrocatalytic CO${}_{2}$ reduction process of both Ti and V atoms is high and cannot compete with HER. Therefore, we only consider the formation of the *OCHO intermediate step in the first step of the protonation process. Figure 8a shows Ti-THQ and Figure 8b shows the step diagram of the V-THQ electrocatalytic CO${}_{2}$ reduction process. The two atom-catalyzed processes are similar, and the formation of *OCHO intermediates is an exothermic reaction as a free energy drop process. The further protonation process to form *OCHOH also shows a free energy drop and is prone to reaction. After *OCHOH formation, the further protonation process of both atoms has a higher energy barrier for the formation of *OCH than for the formation of *CHO, and thus, the formation of *OCH intermediates is not considered. In the subsequent *CHO → *OCH_2_ → *OCH_3_ → *O/*CH_3_OH → *OH are both free energy decreasing processes, but in the final 8e process they are both free energy increasing processes, 0.473 eV (see Appendix A) and 0.241 eV (see Appendix A), respectively. Therefore, the main products of both Ti and V atomic electrocatalytic CO${}_{2}$ reduction are CH${}_{4}$, and the rate decisive steps are *OCHOH + H^+^ + e^−^) → *CHO + H_2_O, but the corresponding limiting potentials are different, with 1.043 eV for Ti-THQ and 0.663 eV for V-THQ. It is obvious that the electrocatalytic performance of V-THQ is due to Ti-THQ.

Although the adsorption energy of Cr-THQ for C${}_{1}$ products behaves in line with Ti and V, and only CH${}_{4}$ can be obtained by desorption from the catalyst surface, the Gibbs free energy of Cr-THQ for the *COOH/*OCHO formation in the first protonation step is below that for the formation of *H according to Figure 4, which shows a good inhibition of hydrogen precipitation reaction. Figure 8c shows the free energy step diagram of the Cr-THQ electrocatalytic CO${}_{2}$ reduction process. Appendix A shows the Cr-THQ electrocatalytic CO${}_{2}$ reduction protonation steps and the corresponding free energy changes. In the first step, protonation to form *COOH/*OCHO consists of free energy reduction processes, and the subsequent protonation to form *CO/*OCHOH also consists of free energy reduction processes. In the next step, *CO/*OCHOH → *COH needs to cross a higher energy barrier of 1.488 eV and 1.622 eV, respectively (see Appendix A), while the energy barrier for the formation of *CHO is lower. Thus, in this step, we only consider the *CHO intermediate, where the energy barrier of the *CO → *CHO process is 0.211 eV, which is slightly lower than that of *OCHOH → *CHO at 0.254 eV (Appendix A). In the subsequent protonation steps, almost all of them are free energy reduction processes, except that the energy barrier of 0.032 eV needs to be crossed in the 8e process. Therefore, we believe that the main product of Cr-THQ electrocatalytic CO${}_{2}$ reduction is CH${}_{4}$, and the decisive step is: *CO + H_2_O + H^+^ + e^−^) → *CHO + H_2_O, with the corresponding limiting potential of 0.211 eV.

According to Figure 4, the energy barrier to be crossed in the first step of protonation to form *OCHO in the electrocatalytic CO${}_{2}$ reduction process of both Mn and Fe atoms is high and cannot compete with HER. Therefore, we only consider the formation of the *COOH intermediate step in the first protonation step. Figure 8d shows Mn-THQ, and Figure 8e shows the step diagram of the Fe-THQ electrocatalytic CO${}_{2}$ reduction process. Appendix A show the Mn-THQ and Fe-THQ electrocatalytic CO${}_{2}$ reduction protonation steps and the corresponding free energy changes, respectively. The two atom-catalyzed processes are similar in that the formation of the *COOH intermediate is an exothermic reaction with decreasing free energy. The further protonation process to form *CO also exhibits a free energy drop and is readily reactive. After *CO formation, both atomic further protonation processes exhibit a higher energy barrier for the formation of *COH than for the formation of *CHO, and thus, the formation of *COH intermediates is not considered. After the formation of *OCH${}_{2}$ intermediates, both catalysts are relatively difficult to desorb to generate HCHO, and the energy barriers to be overcome for Mn-THQ and Fe-THQ are 1.105 eV (see Appendix A) and 0.716 eV (see Appendix A), respectively, so that HCHO need not be considered as a product to desorb. Therefore, the main products of both Mn and Fe monoatoms are CH${}_{4}$. The rate-determining steps are *CO + H_2_O + H^+^ + e^−^) → *CHO + H_2_O and *CHO + H_2_O + H^+^ + e^−^) → *OCH_2_ + H_2_O, corresponding to the ultimate potentials are 0.145 V and 0.147 V, respectively.

The free energy step diagram for each intermediate step of the Zn-THQ electrocatalytic CO${}_{2}$ reduction is shown in Figure 8f. Appendix A shows the reaction equations of each protonation step of Zn-THQ and the corresponding Gibbs free energy changes. Figure 8f shows that the first step of protonation to form *COOH/*OCHO are both free energy step-up processes, which need to overcome the barriers of 0.783 eV and 0.307 eV, respectively. The subsequent formation of *CO/*OCHOH consists of free-energy lowering processes. However, among the subsequent steps, only the *CO → *CHO step requires a smaller energy barrier to be overcome and is the main step that occurs. Unfortunately, the energy barrier that needs to be overcome for the HCOOH desorption step is 0.841 eV (Appendix A), which is higher than the energy barrier of 0.783 eV for the formation of *COOH, and thus, the first step of protonation to form *OCHO need not be considered. The main product of Zn-THQ electrocatalytic CO${}_{2}$ reduction is CH${}_{4}$, and the reaction path is: * + CO_2_ → *COOH → *CO → *CHO → *OCH_2_ → *OCH_3_ → *CH_3_OH → *OH → * + H_2_O. The reaction rate-determining step is * + CO_2_ + H^+^ + e^−^) → *COOH, which corresponds to a limiting potential of 0.783 V.

#### 3.5.4. Carbon Monoxide Is the Main Catalytic Product

Figure 9 shows the Cu-THQ free energy steps for each step of the protonated carbon dioxide reduction. The reaction equations and the corresponding Gibbs free energy variations for each protonation step are shown in Appendix A. After carbon dioxide is attracted to the Cu-HQ surface, the protonation reduction reaction proceeds in the presence of an external voltage. It can be seen in Figure 9 that the first step of protonation to form *COOH requires overcoming a lower energy barrier, and therefore, the formation of *COOH intermediates dominates. The second step of protonation to form *CO is a free energy reduction process. After *CO formation, the formation of *CHO/*COH by possible further protonation requires overcoming an energetic barrier of 0.767 eV/1.897 eV (Appendix A), whereas the step where CO desorption occurs to obtain the product only requires the overcoming of an energetic barrier of 0.65 eV (Appendix A). Therefore, the reaction *CO + H_2_O → * + CO + H_2_O dominates to obtain the product of CO and terminate the reaction. The reaction path of the whole process is * + CO_2_ → *COOH → *CO → CO. The step that determines the rate is * + CO_2_ + H^+^ + e^−^) → *COOH, which corresponds to a potential limit of −0.888 V.

### 3.6. Electronic Structure Analysis

Following the Gibbs free energy change for each intermediate step in Section 3.5, we discuss the rate-determining step, the limiting potential, and the corresponding major products for each catalyst. Among the ten 3D transition metal elements, the Ni-HTQ catalysts for electrocatalytic CO${}_{2}$ in solution indicate that the HER reaction is absolutely dominant. The primary reduction product of Sc-THQ is HCOOH. The main product of Co-THQ is HCHO. The main product of Cu-THQ is CO. The main products of the six elements Ti, V, Cr, Mn, Fe, and Zn are CH${}_{4}$.

The rate-determining steps, limiting potentials and overpotentials of these nine metal-catalyzed processes are given in Table 3. Ti-HQ has the highest limiting and overpotential of 1.043 V and 1.212 V, whereas the overpotentials of the other monolayer catalysts are between 0.172 and 0.952 V, which is comparable to the surface of the most active step Cu(211) (η = 0.77 V) and the surface of the metal Pt(111) (η = 0.46 V) [44]. Thus, our results of theory suggest a very prospective single-atom catalyst for electrocatalytic CO${}_{2}$ abatement.

Organometallic catalysts with metal–ligand bonding theory suggest that the interaction between catalyst and intermediate is mainly σ- and π-bonds. Figure 10 shows a clear overlap between the 3D orbits of metallic atoms and the 2p orbits of O or C atoms in the determining step intermediates (*OCHOH,*CO,*CHO or *COOH), either spinning up or spinning down, which indicates a good interaction between the TM-HQ monolayer and the intermediate. All these TM-THQs have non-zero electronic density of states at the Fermi energy level and exhibit metallicity, with the most obvious metallicity of Cr and Mn. However, the overlap effect of the 3D and 2P orbitals in Figure 10a–c is better than that of the 3D and 2P orbitals in Figure 10e–g). This indicates a stronger interaction of Sc, Ti and V with the respective intermediates than that of Cr, Mn and Fe. The stronger the interaction, the more stable the adsorption intermediate system is and the higher the energy barrier that needs to be overcome to ensure the catalytic reaction, which leads to a larger increase in the free energy of the decisive step of the Ti-THQ-catalyzed CO${}_{2}$ reduction, resulting in a more negative limiting potential for the reduction reaction. As can be seen in Table 3, the limiting potentials $U_{L}$ for the Sc-THQ, Ti-THQ, and V-THQ electrocatalytic CO${}_{2}$ reduction are 0.816 eV, 1.043 eV, and 0.633 eV, respectively, which are higher than those of the Cr, Mn, and Fe catalysts. This is very consistent with the results of PDOS. In addition, the limiting potential of Co is 0.242 V, which is lower than that of Cu and Zn. The overlap effect of 3D orbitals and 2P orbitals in Figure 10g is better than that in Figure 10h,i, which indicates that Co has stronger interaction with *COOH than Cu and Zn and exhibits a more stable structure. Thus, the energy barrier to be crossed for the formation of *COOH by the first step of protonation after CO${}_{2}$ adsorption on the surface of Co-THQ is lower than that of Cu and Zn.

## 4. Conclusions

In the present work, electrocatalytic CO${}_{2}$ reduction reactions with single-atom catalysts creating from transition metal-tetrahydroxyquinone two-dimensional ligand networking materials are investigated. Density functional theory measurements show that metallic atom binding energies to THQ are large enough for 10 transition metal TM-THQ monolayers ranging from Sc to Zn to allow the stable dispersion of metal atoms in the THQ monolayer.The Ni-THQ catalyst does not compete for HER reactions during electrocatalytic CO${}_{2}$ reduction. The major reduction product of Sc-THQ is HCOOH, and the major product of Co-THQ is HCHO. The main product of Cu-THQ is CO. The main product of six elements, Ti, V, Cr, Mn, Fe and Zn, is CH${}_{4}$. Ti-THQ has the highest limiting potential and overpotential of 1.043 V and 1.212 V, respectively, while the overpotential of other monolayer catalysts ranges from 0.172 to 0.952 V, all of which are relatively low. Therefore, we predict that the TM-THQ monolayer will show great catalysis activities in the electocatalytic CO${}_{2}$ reduction process, making it a promising electrocatalyst for CO${}_{2}$ reduction.

## Figures and Tables

**Figure 1 nanomaterials-12-04049-f001:**
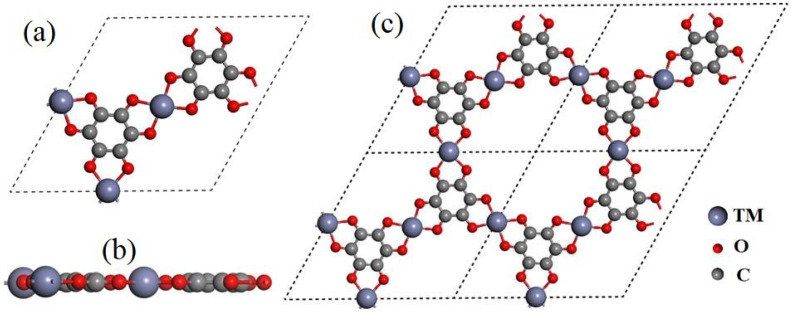
Structural diagram of TM-THQ single layer. (**a**) The top view of the single layer, (**b**) the side view of the single layer, and (**c**) the top view of the 2 × 2 super unit.

**Figure 2 nanomaterials-12-04049-f002:**
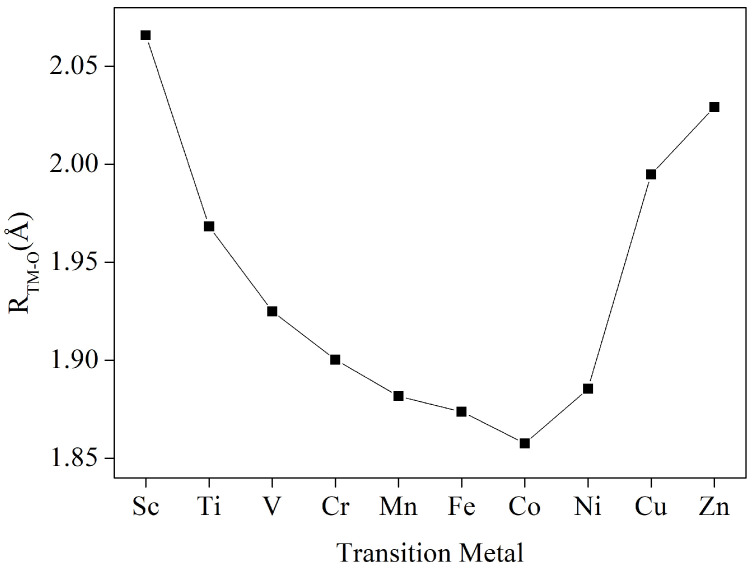
Bond length of the metal atom to the nearest O atom in the TM-THQ monolayer.

**Figure 3 nanomaterials-12-04049-f003:**
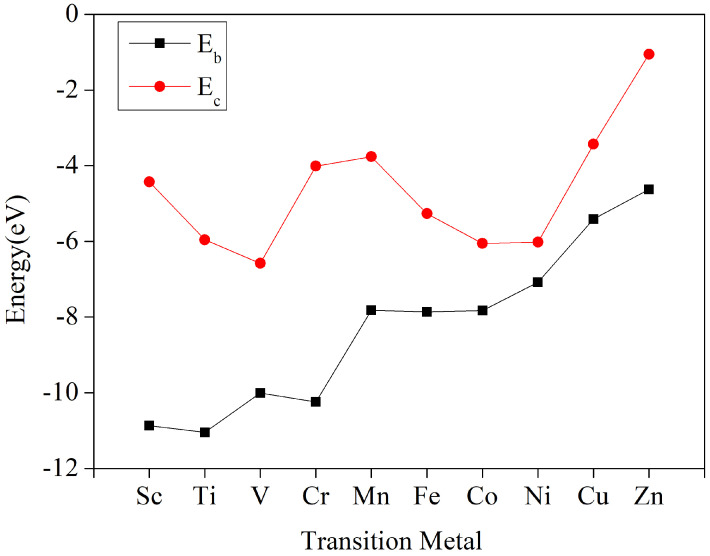
$E_{c}$ is the cohesion energy of the bulk metal, $E_{b}$ is the binding energy between the transition metal atoms and the tetrahydroxybenzoquinone monolayer.

**Figure 4 nanomaterials-12-04049-f004:**
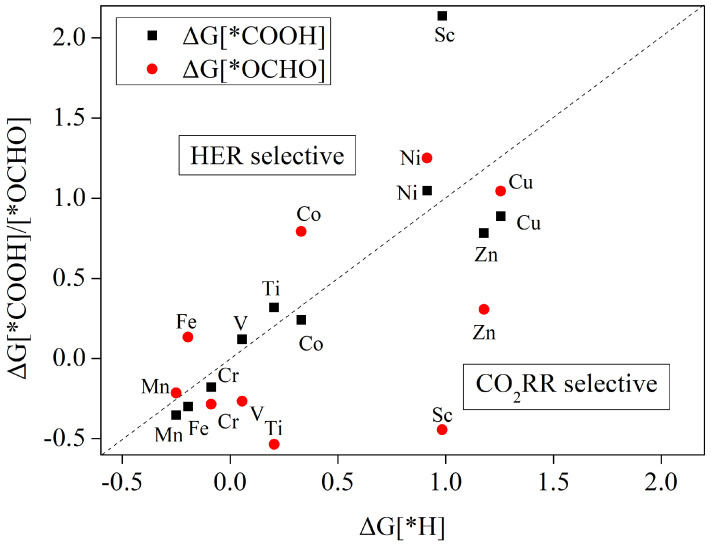
CO${}_{2}$RR versus HER at the first step of protonation Gibbs free energy on the TM-THQ monolayer surface. The catalyst is below the dashed line indicating good CO${}_{2}$RR selectivity.

**Figure 5 nanomaterials-12-04049-f005:**
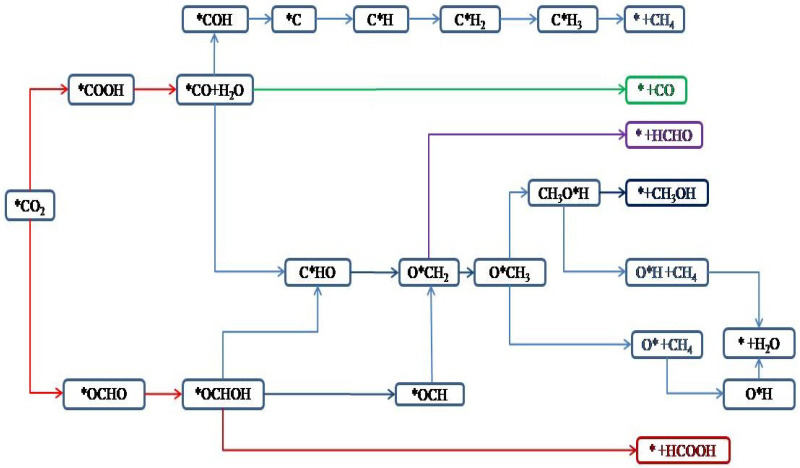
Flow chart of electrocatalytic CO${}_{2}$ reduction to C${}_{1}$ product scheme; red is the final product [43].

**Figure 6 nanomaterials-12-04049-f006:**
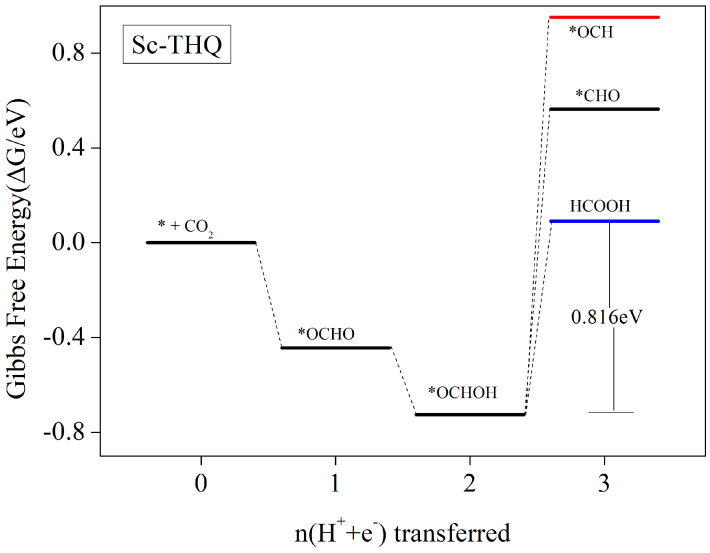
Gibbs free energy profile of CO${}_{2}$RR following the path of most favorable Zn-HAB at the zero potentiation. CO${}_{2}$ molecules in the gaseous phase is given a free energy of zero on the clean catalytic surface.

**Figure 7 nanomaterials-12-04049-f007:**
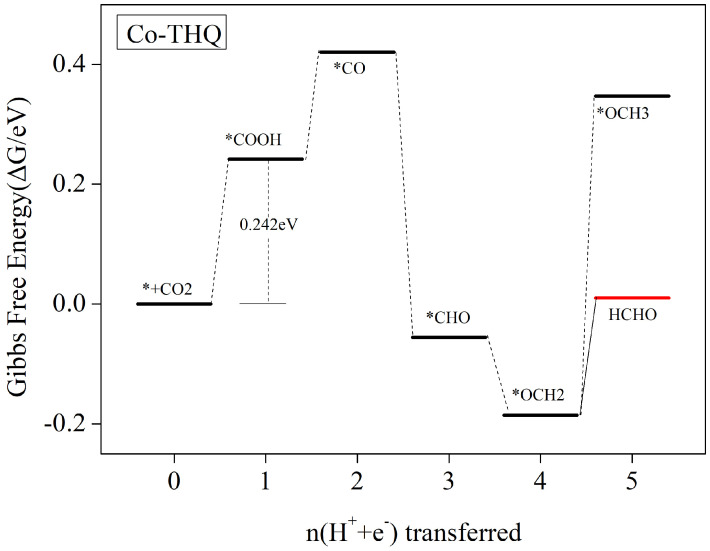
Gibbs free energy profile of CO${}_{2}$RR at zero potential along the most favorable route of Co-THQ. In the aseous stage, the free energy of a CO${}_{2}$ molecule with a clean catalytic surface is zero.

**Figure 8 nanomaterials-12-04049-f008:**
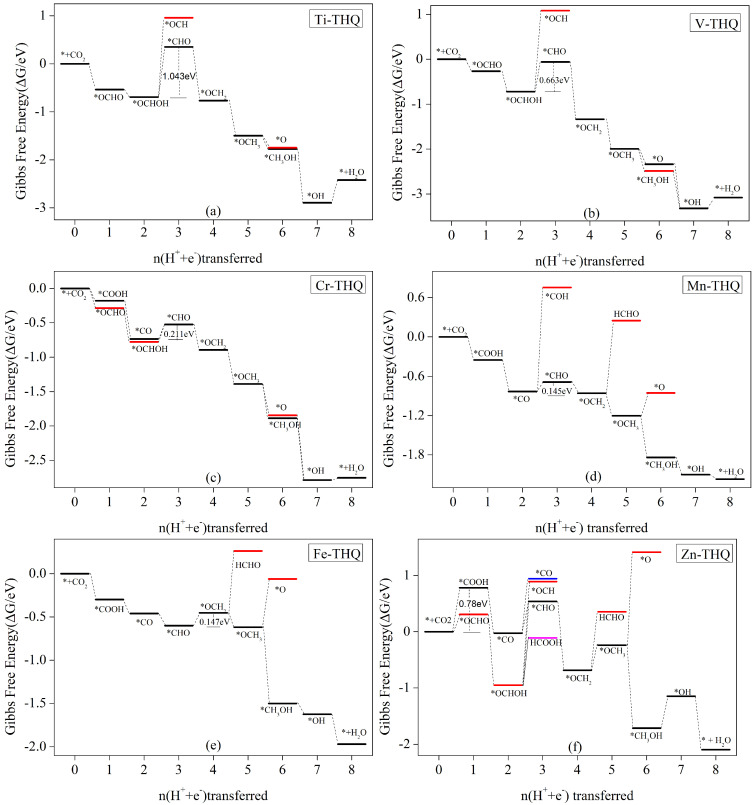
Gibbs free energy curves for (**a**) Ti-THQ, (**b**) V-THQ, (**c**) Cr-THQ, (**d**) Mn-THQ, (**e**) Fe-THQ, and (**f**) Zn-THQ, at zero potential, along the most favorable path of the CO${}_{2}$RR. The free energy zero point was set as the carbon dioxide molecule in the gaseous phase with a squeaky clean surface for the catalyst.

**Figure 9 nanomaterials-12-04049-f009:**
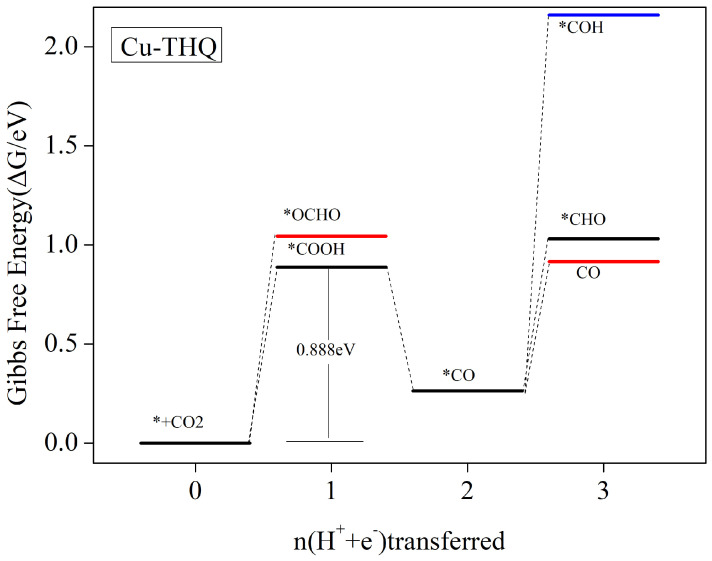
The Gibbs free-energy curve of CO${}_{2}$RR along the most favorable route of Cu-HQ at zero potential. The CO${}_{2}$ molecules with clean catalytic surfaces in the gas phase are assigned a free energy of zero.

**Figure 10 nanomaterials-12-04049-f010:**
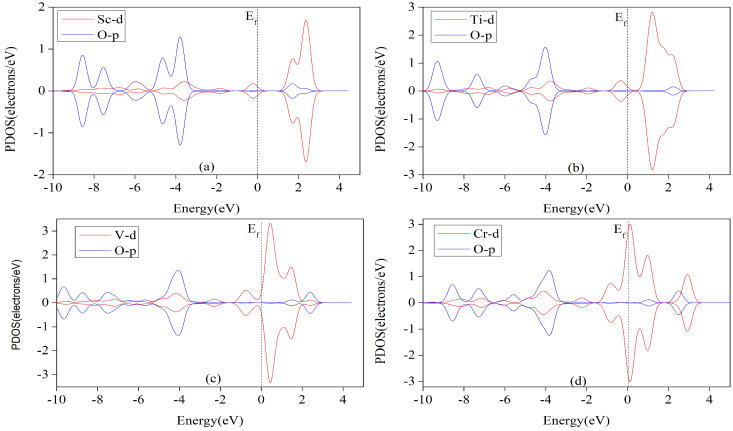
Predicted partial density of states of *OCHOH adsorbed on Sc, Ti, V, and Cr (**a**–**d**). *CO adsorbs on Mn (**e**), *CHO adsorbs on Fe (**f**), and *COOH adsorbs on Co, Cu, and Zn (**g**–**i**). The dashed lines indicate Fermi levels. The red, blue and green wires represent the 3D orbitals of metallic atoms, the 2P orbitals of the oxygen atoms and the 2P orbitals of the carbon atoms, respectively.

**Table 1 nanomaterials-12-04049-t001:** Hirshfeld charges on the metal atom $Q_{T}M$ and oxygen atom $Q_{O}$ of the 10 TM-THQ system, the spin of the metal atom and the length of the TM-O.

TM-THQ	Q${}_{\mathrm{TM}}$/e	Spin-TM	Q${}_{O}$/e	R${}_{\mathrm{TM}-O}$/Å
Sc	0.879	0.000	−0.260	2.066
Ti	0.745	0.000	−0.230	1.968
V	0.607	0.000	−0.204	1.925
Cr	0.659	−2.977	−0.227	1.900
Mn	0.403	0.000	−0.173	1.881
Fe	0.329	0.000	−0.162	1.874
Co	0.283	0.000	−0.156	1.858
Ni	0.275	0.000	−0.158	1.885
Cu	0.480	0.000	−0.210	1.995
Zn	0.582	0.000	−0.226	2.029

**Table 2 nanomaterials-12-04049-t002:** Adsorption energies ($E_{ads}/eV$) with different reduced carbon dioxide products.

TM-THQ	CO	HCOOH	HCHO	CH${}_{3}$OH	CH${}_{4}$
Sc	−0.681	−0.952	−0.135	−0.879	−0.043
Ti	−1.128	−1.026	−1.108	−1.160	−0.176
V	−1.454	−1.196	−1.217	−1.161	−0.174
Cr	−1.491	−1.047	−1.044	−1.031	−0.204
Mn	−1.624	−0.840	−0.631	−1.099	−0.368
Fe	−1.323	−1.367	−0.745	−1.054	−0.415
Co	−0.557	−0.406	−0.351	−0.499	−0.138
Ni	−0.314	−0.311	−0.266	−0.242	−0.096
Cu	−0.539	−0.637	−0.448	−0.611	−0.313
Zn	−0.918	−0.8597	−0.899	−1.013	−0.619

**Table 3 nanomaterials-12-04049-t003:** TM-HTQ is used as a rate-determining step for electrocatalytic CO${}_{2}$ reduction, limiting potential ($U_{L}$/V), overpotential (η/V).

TM-THQ	PDS	$U_{L}$	η
Sc	*OCHOH → * + HCOOH	−0.816	0.566
Ti	*OCHOH + H^+^ + e^−^) → *CHO + H_2_O	−1.043	1.212
V	*OCHOH + H^+^ + e^−^) → *CHO + H_2_O	−0.663	0.832
Cr	*OCHOH + H^+^ + e^−^) → *CHO + H_2_O	−0.211	0.38
Mn	*CO + H_2_O + H^+^ + e^−^) → *CHO + H_2_O	−0.145	0.314
Fe	*CHO + H_2_O + H^+^ + e^−^) → *OCH_2_ + H_2_O	−0.147	0.316
Co	* + CO_2_ + H^+^ + e^−^) → *COOH	−0.242	0.172
Cu	* + CO_2_ + H^+^ + e^−^) → *COOH	−0.888	0.782
Zn	* + CO_2_ + H^+^ + e^−^) → *COOH	−0.783	0.952

## Data Availability

The data presented in this study are available in Appendix A.

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
