# Peer review of "Two-Dimensional (2D) TM-Tetrahydroxyquinone Metal–Organic Framework for Selective CO2 Electrocatalysis: A DFT Investigation"

_nanomaterials, 2022, doi:10.3390/nano12224049_

Round 1

Reviewer 1 Report

CO2RR is a promising technology to transform CO2 into useful chemicals and MOF’s have a tailorable nature which makes them appropriate for the design of new catalysts. In the present paper the authors aim at providing, for the TM-THQ system, systematic information on how the structural parameters control the performance of this particular type of MOF’s.

One of the key topics in CO2RR is the selectivity of this process in comparison with the competing HER process and the paper provides results enabling to select the most promising TMs.

The competition between formate vs CO formation routes is also well addressed and extended to other possible C1 chemicals, such as methane and the best metal candidates are indicated for each route.

In my opinion this work can be published after minor corrections of typos the most important is the lack of a legend in table 1.

Author Response

Dear reviewer,

We would like to thank the comments from you, which much helpful for revision of our manuscript (#nanomaterials-2030873 ). We revised our manuscript accordingly and the corrections are listed below:

  1. In my opinion this work can be published after minor corrections of typos the most important is the lack of a legend in table 1.

Thank you very much for your suggestion. We have made the following changes.

 (1) To get it right, we've replaced the first sentence of the conclusion with “In the present work, the two-dimensional transition metal-tetrahydroxyquinone electrocatalytic carbon dioxide reduction reaction are investigated.”

(2)We have used the words generation, obtained, and production to replace the formation in the “ TM-THQ single layer for CO2RR and HER specificity” part.

(3) We have added a plot of the bond lengths of the metal atom to the nearest O atom in the TM-THQ monolayer (see Figure 2) as a visual supplement to Table 1.

Thank you again for your positive comments and valuable suggestions to improve the quality of our manuscript.

Best wishes.

Xianshi Zeng

Nanchang University

Reviewer 2 Report

Tetrahydroxyquinone-based 2D MOF materials are promising systems for electrocatalytic CO2 reduction. So, the rational design of these electrocatalysts is strongly needed. So, the paper by Y. Wen is interesting from this point of view. In particular, the possibility to predict main products in CO2RR is interesting for different electrocatalyst M-THQ depending on metal is important. This paper may be accepted for publication in Nanomaterials after minor revision by addressing the following issues.

1. How 2D structure of the selected electrocatalytic materials M-THQ is taken into account in modeling?

2. The trends of binding energy increasing along with specific metal position, which are presented on the Figure 2 should be discussed in detail including the reasons for the fluctuations for these trends for Eb and Ec.

3. Conclusion. Line 419. What is meaning for “transition metal-tetrahydroxyquinone two-dimensional ligand networking materials”?

4. The style correction is needed. For instance, Lines 197-199. The word “formation” is used 7 times.

Author Response

Dear reviewer,

We would like to thank the comments from you, which much helpful for revision of our manuscript (#nanomaterials-2030873 ). We revised our manuscript accordingly and the corrections are listed below:

  1. How 2D structure of the selected electrocatalytic materials M-THQ is taken into account in modeling?

Thank you very much for your suggestion. Cu-THQ has been synthesized experimentally, reference is: Leily Majidi, Alireza Ahmadiparidari, et al 2D Copper Tetrahydroxyquinone Conductive Metal–Organic Framework for Selective CO2 Electrocatalysis at Low Overpotentials. Advanced Materials, 33, 2004393. Based on the Cu-THQ model provided in the literature, we constructed a series of TM-THQ models for prediction.

  1. The trends of binding energy increasing along with specific metal position, which are presented on the Figure 2 should be discussed in detail including the reasons for the fluctuations for these trends for Eband Ec.

Thank you very much for your suggestion. We have added a discussion of Ec fluctuations as follows:The reason for the irregular fluctuations of the cohesion energy of the ten metals is that the ten transition metals belong to different crystal systems, among which Sc, Ti, Co, and Zn belong to hexagonal structure; V, Cr, Fe, Ni, and Cu belong to cubic crystal system structure, and Mn belongs to trigonal crystal system structure. Different crystal systems have different atomic arrangements, which affect different ground state energies and thus exhibit irregular fluctuations.

  1. Conclusion. Line 419. What is meaning for “transition metal-tetrahydroxyquinone two-dimensional ligand networking materials”?

Thank you very much for your suggestion. To get it right, we've replaced the first sentence of the conclusion with “In the present work, the two-dimensional transition metal-tetrahydroxyquinone electrocatalytic carbon dioxide reduction reaction are investigated.”

  1. The style correction is needed. For instance, Lines 197-199. The word “formation” is used 7 times.

Thank you very much for your suggestion. We have used the words generation, obtained, and production to replace the formation.

Thank you again for your positive comments and valuable suggestions to improve the quality of our manuscript.

Best wishes.

Xianshi Zeng

Nanchang University
